# A Differential-Geometric Approach to Quantum Ignorance Consistent with Entropic Properties of Statistical Mechanics

**DOI:** 10.3390/e25050788

**Published:** 2023-05-12

**Authors:** Shannon Ray, Paul M. Alsing, Carlo Cafaro, H S. Jacinto

**Affiliations:** 1Air Force Research Laboratory, Rome, NY 13441, USA; paul.alsing@us.af.mil (P.M.A.); h.jacinto@afrl.af.mil (H.S.J.); 2Griffiss Institute, Rome, NY 13441, USA; 3Department of Mathematics and Physics, SUNY Polytechnic Institute, Albany, NY 12203, USA; cafaroc@sunypoly.edu

**Keywords:** differential geometry, entanglement entropy, quantum information, statistical physics, coarse-graining, many-body systems, ignorance, thermalization, Lie groups

## Abstract

In this paper, we construct the metric tensor and volume for the manifold of purifications associated with an arbitrary reduced density operator $\rho_{S}$. We also define a quantum coarse-graining (CG) to study the volume where macrostates are the manifolds of purifications, which we call surfaces of ignorance (SOI), and microstates are the purifications of $\rho_{S}$. In this context, the volume functions as a multiplicity of the macrostates that quantifies the amount of information missing from $\rho_{S}$. Using examples where the SOI are generated using representations of SU(2), SO(3), and SO(N), we show two features of the CG: (1) A system beginning in an atypical macrostate of smaller volume evolves to macrostates of greater volume until it reaches the equilibrium macrostate in a process in which the system and environment become strictly more entangled, and (2) the equilibrium macrostate takes up the vast majority of the coarse-grained space especially as the dimension of the total system becomes large. Here, the equilibrium macrostate corresponds to a maximum entanglement between the system and the environment. To demonstrate feature (1) for the examples considered, we show that the volume behaves like the von Neumann entropy in that it is zero for pure states, maximal for maximally mixed states, and is a concave function with respect to the purity of $\rho_{S}$. These two features are essential to typicality arguments regarding thermalization and Boltzmann’s original CG.

## 1. Introduction

In this paper, we introduce a new volume associated with an arbitrary density operator $\rho_{S}$ that quantifies the ignorance or information missing from $\rho_{S}$ relative to purifications that can generate it. To compute this volume, we generate all purifications of $\rho_{S}$ using the method in Section 9.2.3 (Uhlmann Fidelity) of [1] and construct the metric tensor of the manifold of purifications. The determinant of the metric tensor gives a volume element which is integrated to compute volumes. We then study these volumes by presenting examples for systems whose purifications are generated using unitaries that represent Lie groups SU(2), SO(3), and SO(N). Because these volumes are related to the amount of information missing in $\rho_{S}$, we denote the manifolds of purifications as surfaces of ignorance (SOI).

To study the physical properties of our volume, we formulate the SOI as macrostates of an entanglement-based quantum coarse-graining (CG) where microstates are the purifications that belong to each SOI; density operators $\rho_{S}$ are also the macrostates since there is a one-to-one correspondence between them and the SOI. The reason for choosing this context is that the entanglement entropy has been shown to be closely related to thermal entropy in certain regimes [2,3,4,5], and $\rho_{S}$ can be treated as a reduced density operator, $\rho_{S}={\mathrm{Tr}}_{E}[|\psi_{ES}\rangle \langle \psi_{ES}|]$, of a closed composite system $|\psi_{ES}\rangle$. Since $\rho_{S}$ is a reduced density operator of a pure composite system, the von Neumann entropy, $S_{VN}$, of $\rho_{S}$ is the entanglement entropy between the system, *S*, and the environment, *E*. This implies that an increase in volume during an entangling process relates to a loss of information from *S* to *E* that is reminiscent of an information-based thermalization. Although the entanglement entropy is related to thermal entropy, as stated in [6], “it still primarily measures the information exchange rather than heat exchange”. For this reason, our analysis is not a study of thermalization. Instead, it is an exploration of the SOI and their volumes in the context of “thermalization” as it relates to information exchange/entanglement. Our choice to use CG to study our volume is also justified since using reduced density operators as coarse representations of composite systems is common within the literature [7,8,9,10,11,12].

With this context in mind, there are two features of Boltzmann’s original CG [13] (see Figure 1) that we demonstrate in the examples of our entanglement coarse-graining (ECG). These features are the following: (1) a system beginning in an atypical macrostate of smaller volume evolves to macrostates of greater volume until it reaches the equilibrium macrostate in a process in which the system and environment become strictly more entangled, and (2) the equilibrium macrostate takes up the vast majority of the coarse-grained space especially as the dimension of the total system becomes large.

These features are the basis of typicality arguments for understanding the thermalization of both classical and quantum closed systems [15,16].

Quantum mechanically, $S_{VN}(|\psi_{ES}\rangle \langle \psi_{ES}|)=0$ for all evolutions of $|\psi_{ES}\rangle$ in the space of purifications. Therefore, it is common practice [17,18,19] to demarcate the space of purifications into disjoint sets, or macrostates, for which thermal entropies are defined. For the ECG, the SOI provide this demarcation and their volumes are treated as the multiplicity of a strictly information-based “thermal” entropy. It is not our goal to define a quantum Boltzmann entropy, and we are not interested in studying energy or dynamics. Instead, we only analyze volumes and use a purely kinematic approach afforded to us by the ECG. This makes our approach similar to Boltzmann’s original analysis and that in [20], which studied the foundations of statistical mechanics in terms of entanglement.

To demonstrate feature (1) for the examples considered, we must show that the volume behaves like $S_{VN}\left(\rho_{S}\right)$ in that it is zero for pure state, maximal for maximally mixed states, and is a concave function with respect to the purity of $\rho_{S}$. This implies that each SOI has a unique entanglement entropy associated with it. It is also consistent with thermalization as described by Boltzmann’s CG where the total system monotonically evolves between macrostates of lesser volume to macrostates of greater volume until it reaches the most typical macrostate that occupies the vast majority of the coarse-grained space.

In studies that use typicality arguments to understand thermalization, the equilibrium macrostate is defined as the largest macrostate that occupies the vast majority of the coarse-grained space [17,18,19]. This also defines the equilibrium macrostate for the ECG, but it has the additional trait that its microstates have maximal entanglement between *S* and *E*; this is synonymous with $\rho_{S}$ being maximally mixed. Therefore, to demonstrate feature (2), we study the average von Neumann entropy of each macrostate belonging to the ECG generated by SO(3) and show that the majority of the coarse-grained space is occupied by the macrostates with maximum or near-maximum entanglement entropy. We further show, using SO(N), that the average normalized von Neumann entropy of at least 99.99% of the coarse-grained space tends toward one (maximally mixed) as *N* becomes large. The use of 99.99% as a representative value for the vast majority of the coarse-grained space is commonly used in the literature [14,19,21,22].

The final context in which we relate our volume to the multiplicity of a Boltzmann-like entropy is discussed in section IIC of [23] and provided by [24]. In that analysis, Brillouin used the Maxwell demon gedanken to connect negentropy [25,26] (information) to the Boltzmann entropy. More specifically, he showed that the greater the multiplicity of microstates that are consistent with macrodata, the less information one has about the total system. In our case, the negentropy is defined as
(1)$$I=S_{VN}^{max}-S_{VN}\left(\rho_{S}\right)$$
where $S_{VN}^{max}$ is the von Neumann entropy of the maximally mixed density operator, and $\rho_{S}$ contains the remaining information of $|\psi_{ES}\rangle$ after the partial trace has been taken. This means if one only has the macrodata contained in $\rho_{S}$, they no longer know which purification, i.e., microstate, completes the missing information of $\rho_{S}$. Therefore, the greater the volume of the SOI, the more purifications there are, which implies one is less likely to successfully guess at random the actual pure state that produced $\rho_{S}$. Furthermore, this guess must be random because to use anything other than a maximally mixed distribution on the purifications of $\rho_{S}$ would, as stated by Jaynes [27], “amount to an arbitrary assumption of information which by hypothesis we do not have”.

The paper is structured as follows. In Section 2, we construct the metric components and volume of the SOI. In Section 3, we study the volume in the context of the ECG using unitaries representing Lie groups SO(3), SU(2), and SO(N). In Section 4, we generalize the ECG and the metric components of the SOI to include unitary transformations in $H_{S}$. Finally, we conclude in Section 5 with a summary of our results.

## 2. Methods: Entanglement Coarse-Graining and the Surfaces of Ignorance

In this section, we define the macro- and microstates of the ECG and derive the metric components and volume of the SOI.

### 2.1. Macro and Microstates

In the ECG, macrostates are density operators $\rho_{S}$ (as well as the SOI associated with each $\rho_{S}$), and microstates are elements of the set of purifications $F^{\rho_{S}}\equiv \{|{\bar{\Gamma }}_{ES}^{\rho_{S}}\left(\vec{\xi }\right)\rangle \}$ such that
(2)$$\rho_{S}={\mathrm{Tr}}_{E}\left[|{\bar{\Gamma }}_{ES}^{\rho_{S}}\left(\vec{\xi }\right)\rangle \langle {\bar{\Gamma }}_{ES}^{\rho_{S}}\left(\vec{\xi }\right)|\right].$$ The space of the environment, $H_{E}$, is taken as a copy of $H_{S}$ since it is sufficient to generate all purifications of $\rho_{S}$, and $\vec{\xi }$ parameterizes the transformations $U_{E}\left(\vec{\xi }\right)$ that represent the Lie group symmetry of $H_{E}$.

Writing $\rho_{S}$ in its spectral form
(3)$$\rho_{S}=\sum_{i=1}^{N}\lambda^{i}|\lambda_{S}^{i}\rangle \langle \lambda_{S}^{i}|,$$
where *N* is the dimension of $H_{S}$, the macrodata are the eigenvalues $\vec{\lambda }$. For an orthonormal basis $\left\{|\lambda_{S}^{i}\rangle \right\}$ of $H_{S}$, the set of all eigenvalues that satisfy the constraint
(4)$$\sum_{i=1}^{N}\lambda^{i}=1,$$
gives a probability simplex S where each element of S is a valid density operator. The probability simplex is a subspace of the projective space $P(H_{S})$, the latter being defined by all normalized rank-one projectors of $H_{S}$ that are well defined up to U(1) symmetries. Since each $\rho_{S}$ on S has a unique $F^{\rho_{S}}$, there exists a unique ECG of $H_{ES}$ associated with S; this is depicted in Figure 2, which shows an information/entanglement-based “thermalization” process.

To generate $F^{\rho_{S}}$ we follow the prescription given in 9.2.3 of Wilde’s “Quantum Information Theory” [1]. We begin with the canonical purification
(5)$$|\phi_{ES}^{\rho_{S}}\rangle =\left({\hat{1}}_{E}⊗\sqrt{\rho_{S}}\right)|\Gamma_{ES}\rangle$$
in $H_{ES}$, where ${\hat{1}}_{E}$ is the identity operator in $H_{E}$,
(6)$$|\Gamma_{ES}\rangle =\sum_{i=1}^{N}|\lambda_{E}^{i}\rangle |\lambda_{S}^{i}\rangle$$
is the unnormalized Bell state, and $\left\{|\lambda_{E}^{i}\rangle \right\}$ is a copy of $\left\{|\lambda_{S}^{i}\rangle \right\}$ in $H_{E}$. From here, one can access all purifications by applying unitary transformations associated with the symmetries of $H_{E}$ to Equation (Equation 5). This gives
(7)$$|{\bar{\Gamma }}_{ES}^{\rho_{S}}\left(\vec{\xi }\right)\rangle =\left(U_{E}\left(\vec{\xi }\right)⊗{\hat{1}}_{S}\right)|\phi_{ES}^{\rho_{S}}\rangle =\left(U_{E}\left(\vec{\xi }\right)⊗\sqrt{\rho_{S}}\right)|\Gamma_{ES}\rangle .$$

In general, $H_{E}$ need not be a copy of $H_{S}$ since $\rho_{S}$ can be derived from any bipartition of an arbitrary many-body system $|\psi_{ES}\rangle$. Therefore, to generalize the macrostates of the ECG given by Equation (Equation 7) to an arbitrary purification space $H_{\bar{E}S}$ where $H_{\bar{E}}\neq H_{S}$, we use the fact that all purifications of $\rho_{S}$ are unitarily related.

Given the restriction that $dim(\bar{E})\geq N$, the ECG of $H_{ES}$ can be extended to $H_{\bar{E}S}$ by
(8)$$|{\bar{\Gamma }}_{\bar{E}S}^{\rho_{S}}\left(\vec{\xi }\right)\rangle =\left(U_{E\to \bar{E}}⊗{\hat{1}}_{S}\right)|{\bar{\Gamma }}_{ES}^{\rho_{S}}\left(\vec{\xi }\right)\rangle$$
where
(9)$$U_{E\to \bar{E}}=\sum_{i=1}^{N}|\lambda_{\bar{E}}^{i}\rangle \langle \lambda_{E}^{i}|$$
and $\left\{|\lambda_{\bar{E}}^{i}\rangle \right\}$ is a complete orthonormal basis of $H_{\bar{E}}$. Since all macrostates of $H_{ES}$ can be extended to macrostates of some larger $H_{\bar{E}S}$, we only need to consider the former to define a general ECG.

### 2.2. Surfaces of Ignorance: Metric Components and Volume

To compute the metric components and volume associated with $F^{\rho_{S}}$, we construct its first fundamental form using a Taylor expansion of Equation (Equation 7). Expanding around parameters ${\vec{\xi }}_{0}$ using $\vec{\xi }$, the displacement vector is given by $d\vec{\xi }=\vec{\xi }-{\vec{\xi }}_{0}$. Taking the first-order Taylor expansion of $|{\bar{\Gamma }}_{ES}^{\rho_{S}}\left(\vec{\xi }\right)\rangle$, and bringing the zeroth order term to the l.h.s, the differential is given by
(10)$$|d\bar{\Gamma }\rangle \equiv |\bar{\Gamma }\left({\vec{\xi }}_{0}+d\vec{\xi }\right)\rangle -|\bar{\Gamma }\left({\vec{\xi }}_{0}\right)\rangle =\sum_{i=1}^{n}|{\bar{\Gamma }}_{,\xi_{i}}\rangle d\xi_{i}$$
where *n* is the number of parameters of the unitary representation of the Lie groups, and $|{\bar{\Gamma }}_{,\xi_{i}}\rangle$ is the partial derivative of $|\bar{\Gamma }\rangle$ with respect to $\xi_{i}$. For the remainder of the paper, superscript $\rho_{S}$ and subscript ES are dropped from $|{\bar{\Gamma }}_{ES}^{\rho_{S}}\left(\vec{\xi }\right)\rangle$ for simplicity of notation. Since we are working in $H_{ES}$, and all of our states are pure, the scalar product is well defined. The components $g_{ij}$ of the metric tensor *g* induced by the scalar product are given by the first fundamental form
(11)$$ds^{2}=\langle d\bar{\Gamma }|d\bar{\Gamma }\rangle =\sum_{i,j=1}^{n}\langle {\bar{\Gamma }}_{,i}|{\bar{\Gamma }}_{,j}\rangle d\xi_{i}d\xi_{j}$$
where $g_{ij}=\langle {\bar{\Gamma }}_{,i}|{\bar{\Gamma }}_{,j}\rangle$. From Equation (Equation 11), the volume element is $dV=\sqrt{\mathrm{Det}[g]}\;d\xi_{1}d\xi_{2}\cdots d\xi_{n}$ and the volume is
(12)$$V=\int_{\xi_{1}}\int_{\xi_{2}}\cdots \int_{\xi_{n}}dV.$$

## 3. Results: Volume Examples

In this section, we give explicit expressions of volumes for the examples considered and compare them to the von Neumann entropy, $S_{VN}=-\sum_{i=1}^{N}\lambda^{i}\log\lambda^{i}$, and the linear entropy, $S_{L}=1-\mathrm{Tr}\left[\rho_{S}^{2}\right]$. We demonstrate feature (1) of Boltzmann’s original CG for SU(2), features (1) and (2) for SO(3), and extend the demonstration of feature (2) for SO(3) in the limit of large *N* using SO(N). However, first, we give the expressions for arbitrary unitary transformations that are used to compute the metric components and volumes for our examples.

### 3.1. Arbitrary N-Dimensional Unitary Transformations

Following the prescription in [28], any arbitrary *N*-dimensional unitary transformation can be written as successive transformations of two-dimensional subspaces. Let $E^{\left(i,j\right)}\left(\phi_{ij},\psi_{ij},\chi_{ij}\right)$ be an arbitrary transformation about the (i,j)-plane. Its components are
(13)$$\begin{array}{ccc} E_{kk}^{\left(i,j\right)} & = & 1\;\;\;\;\;\;k=1,\cdots ,N\;\;\;\;\;\;k\neq i,j\\ E_{ii}^{\left(i,j\right)} & = & e^{i\psi_{ij}}\cos\phi_{ij}\\ E_{ij}^{\left(i,j\right)} & = & e^{i\chi_{ij}}\sin\phi_{ij}\\ E_{ji}^{\left(i,j\right)} & = & -e^{-i\chi_{ij}}\sin\phi_{ij}\\ E_{jj}^{\left(i,j\right)} & = & e^{-i\psi_{ij}}\cos\phi_{ij} \end{array}$$
and zero everywhere else. The superscript indices (i,j) index the 2-D plane about which the transformation is applied, and the subscripts are the nonzero matrix indices. From here, one can construct successive transformations
(14)E1=E(1,2)(ϕ12,ψ12,χ12)E2=E(2,3)(ϕ23,ψ23,0)E(1,3)(ϕ13,ψ13,χ13)... EN−1=E(N−1,N)(ϕN−1,N,ψN−1,N,0)E(N−2,N)(ϕN−2,N,ψN−2,N,0)(15)⋯E(1,N)(ϕ1N,ψ1N,χ1N)
and finally an arbitrary U(N) transformation
(16)$$U=e^{i\alpha }E_{1}E_{2}\;\cdots \;E_{N-1}$$
where $\phi_{ij}\in \left[0,\pi /2\right]$ and $\alpha ,\psi_{ij},\chi_{ij}\in \left[0,2\pi \right]$. With the arbitrary unitaries defined, we now present our examples.

### 3.2. Example: SU(2)

Here, we demonstrate feature (1) for SU(2) by computing the volume and comparing it to the von Neumann and linear entropies. We do not attempt to demonstrate feature (2) since it is a feature that manifests for large systems and here, the composite system is only four-dimensional.

From Equation (16), the unitaries of SU(2) are given by
(17)$$U\left(\phi ,\psi ,\chi \right)=\left[\begin{array}{ccc} e^{i\psi }\cos\phi & \; & e^{i\chi }\sin\phi \\ -e^{-i\chi }\sin\phi & \; & e^{-i\psi }\cos\phi \end{array}\right]$$
where N=2, α=0, ψ,χ∈[0,2π], ϕ∈[0,π/2], and the subscript 12 in the angles is dropped since the example is only two-dimensional. Computing the metric components directly, the nonzero values of the metric are
(18)$$\begin{array}{ccc} g_{\phi \phi } & = & \lambda^{1}+\lambda^{2} \end{array}$$
(19)$$\begin{array}{ccc} g_{\psi \psi } & = & \left(\lambda^{1}+\lambda^{2}\right)\cos^{2}\phi \end{array}$$
(20)$$\begin{array}{ccc} g_{\chi \chi } & = & \left(\lambda^{1}+\lambda^{2}\right)\sin^{2}\phi \end{array}$$
(21)$$\begin{array}{ccc} g_{\phi \psi } & = & g_{\phi \chi }=i\left(\lambda^{1}-\lambda^{2}\right)\cos\phi \sin\phi . \end{array}$$ Taking the $\sqrt{\mathrm{Det}(g)}$ and substituting $\lambda^{2}=1-\lambda^{1}$ gives
(22)$$dV_{SU(2)}=\sqrt{\lambda^{1}\left(1-\lambda^{1}\right)}\sin2\phi \;d\phi d\psi d\chi$$
and integrating over {ϕ,ψ,χ} gives
(23)$$V_{SU(2)}=4\pi^{2}\sqrt{\lambda^{1}\left(1-\lambda^{1}\right)}=4\pi^{2}\sqrt{S_{L}/2}$$
where $\lambda^{2}=1-\lambda^{1}=\frac{1}{2}\left[1+\sqrt{2\mathrm{Tr}[\rho^{2}]-1}\right]$.

We compare the normalized volume, $V_{SU(2)}^{\mathrm{norm}}$, with the normalized von Neumann entropy, $S_{VN}^{\mathrm{norm}}$, and normalized linear entropy, $S_{L}^{\mathrm{norm}}$, in Figure 3. Each volume/entropy is normalized with respect to their maximum values so that they take values on the interval [0,1]. It is shown that all three functions are zero on pure states, maximal on maximally mixed states, and are concave function with respect to the purity of $\rho_{S}$. This shows that feature (1) is satisfied for this example. In fact, the volume is an upper bound of both entropies. It should also be noted that the behavior of $V_{SU(2)}^{\mathrm{norm}}$ deviates from $S_{VN}^{\mathrm{norm}}$ and $S_{L}^{\mathrm{norm}}$ in that it is flatter near maximally mixed states and steeper near pure states. As we see in Section 3.3, this flatter behavior has implications about feature (2) also being satisfied in that more of the coarse-grained space consists of macrostates with a greater von Neumann entropy. However, one would not expect this feature to be pronounced since the dimension of this example is so low.

### 3.3. Example: SO(3)

This section is broken into two subsections. In Section 3.3.1, we demonstrate feature (1) by computing the volume and comparing it to the linear and von Neumann entropies. In Section 3.3.2, we demonstrate feature (2) by discretizing S to construct an explicit CG. We then compute the average von Neumann entropy of each discrete macrostate and show that a significant majority of the coarse-grained space consists of macrostates with maximum or near-maximum von Neumann entropy, which is consistent with the composite system being maximally entangled.

#### 3.3.1. Computing Volume

From Equation (16), the unitaries associated with SO(3) are given by choosing N=3 and $\alpha =\psi_{ij}=\chi_{ij}=0$ for all *i* and *j*. This leaves parameters $\vec{\xi }=\left(\phi_{12},\phi_{13},\phi_{23}\right)$ where $\phi_{12},\phi_{13},\phi_{23}\in \left[0,\pi /2\right]$. The resulting unitaries are given by
(24)$$\begin{array}{cc} \; & U(\phi_{12},\phi_{13},\phi_{23})=\\ \; & \left[\begin{array}{ccc} \cos\phi_{12}\cos\phi_{13}-\sin\phi_{12}\sin\phi_{13}\sin\phi_{23} & \cos\phi_{23}\sin\phi_{12} & \cos\phi_{12}\sin\phi_{13}+\cos\phi_{13}\sin\phi_{12}\sin\phi_{23}\\ -\cos\phi_{13}\sin\phi_{12}-\cos\phi_{12}\sin\phi_{13}\sin\phi_{23} & \cos\phi_{12}\cos\phi_{23} & -\sin\phi_{12}\sin\phi_{13}+\cos\phi_{12}\cos\phi_{13}\sin\phi_{23}\\ -\cos\phi_{23}\sin\phi_{13} & -\sin\phi_{23} & \cos\phi_{13}\cos\phi_{23} \end{array}\right]. \end{array}$$ Since $U(\phi_{12},\phi_{13},\phi_{23})$ are the unitaries of both $H_{E}$ and $H_{S}$, we use the sublabels *E* and *S* to keep track of which space *U* is acting upon.

Working in the basis of S, $\left\{|\lambda_{S}^{i}\rangle \right\}$ is given by
(25)$$\{|\lambda_{S}^{i}\rangle \}=\left\{\begin{array}{ccc} \left[\begin{array}{c} 1\\ 0\\ 0 \end{array}\right], & \left[\begin{array}{c} 0\\ 1\\ 0 \end{array}\right], & \left[\begin{array}{c} 0\\ 0\\ 1 \end{array}\right] \end{array}\right\}.$$ This gives an explicit form of the unnormalized Bell state given by Equation (Equation 6). From here, all purifications are generated by
(26)$$|\bar{\Gamma }\left(\vec{\xi }\right)\rangle =\sum_{i=1}^{3}\sqrt{\lambda^{i}}U_{E}\left(\vec{\xi }\right)|\lambda_{E}^{i}\rangle ⊗|\lambda_{S}^{i}\rangle .$$ Using Equation (26), the nonzero metric components of $F^{\rho_{S}}\equiv \{|\bar{\Gamma }\left(\vec{\xi }\right)\rangle \}$ are
(27)$$\begin{array}{ccc} g_{\phi_{12}\phi_{12}} & = & \sin^{2}\phi_{23}+\frac{1}{4}\left(\lambda^{1}+\lambda^{2}+3\left(\lambda^{1}+\lambda^{2}\right)\cos2\phi_{23}+2\left(\lambda^{1}-\lambda^{2}\right)\cos2\phi_{13}\sin^{2}\phi_{23}\right) \end{array}$$
(28)$$\begin{array}{ccc} g_{\phi_{13}\phi_{13}} & = & \lambda^{1}+\lambda^{2} \end{array}$$
(29)$$\begin{array}{ccc} g_{\phi_{23}\phi_{23}} & = & \frac{1}{2}\left(2-\lambda^{1}-\lambda^{2}-\left(\lambda^{1}-\lambda^{2}\right)\cos2\phi_{13}\right) \end{array}$$
(30)$$\begin{array}{ccc} g_{\phi_{12}\phi_{13}} & = & g_{\phi_{13}\phi_{12}}=\left(\lambda^{1}+\lambda^{2}\right)\cos\phi_{23} \end{array}$$
(31)$$\begin{array}{ccc} g_{\phi_{12}\phi_{23}} & = & g_{\phi_{23}\phi_{12}}=-\left(\lambda^{1}-\lambda^{2}\right)\cos\phi_{13}\sin\phi_{23}\sin\phi_{13} \end{array}$$
where $g_{\phi_{13}\phi_{23}}=g_{\phi_{23}\phi_{13}}=0$. Taking $\sqrt{\mathrm{Det}(g)}$ gives
(32)$$dV_{SO(3)}=\sqrt{\left(\lambda^{1}+\lambda^{2}\right)\left(\lambda^{1}+\lambda^{3}\right)\left(\lambda^{2}+\lambda^{3}\right)}\cos\phi_{23}\;d\phi_{12}d\phi_{13}d\phi_{23}$$
and integrating over $\vec{\xi }$ gives
(33)$$\begin{array}{ccc} V_{SO(3)} & = & \left(\pi^{2}/4\right)\sqrt{\left(\lambda^{1}+\lambda^{2}\right)\left(\lambda^{1}+\lambda^{3}\right)\left(\lambda^{2}+\lambda^{3}\right)} \end{array}$$
(34)$$\begin{array}{ccc} & \;\;\;\;= & \left(\pi^{2}/4\right)\sqrt{\left(1-\lambda^{1}\right)\left(1-\lambda^{2}\right)\left(\lambda^{1}+\lambda^{2}\right)} \end{array}$$
where the second equality is due to the constraint that the sum of the eigenvalues must equal one.

As for the SU(2) example, we compare the normalized volume, $V_{SO(3)}^{\mathrm{norm}}$, with $S_{VN}^{\mathrm{norm}}$ and $S_{L}^{\mathrm{norm}}$ by plotting them in Figure 4a–d. Here, we see, as was seen for SU(2), that $V_{SO(3)}^{\mathrm{norm}}$ is zero for pure states, maximal on maximally mixed states, and concave with respect to purity, thus satisfying feature (1). Again, as for the SU(2) example, the volume upper bounds $S_{VN}^{\mathrm{norm}}$, as seen in Figure 4d. It also upper bounds $S_{L}^{\mathrm{norm}}$, but we do not show it for the sake of clarity. Notice as well that $V_{SO(3)}^{\mathrm{norm}}$ is flatter near the maximally mixed state and steeper near pure states. This, again, is an indication that it also satisfies feature (2), which we analyze explicitly in Section 3.3.2.

#### 3.3.2. Analyzing the Entanglement Entropy of Macrostates

To demonstrate feature (2) for SO(3), we compute the fraction of S that belongs to each macrostate in the coarse-grained space, $H_{ES}$, and compute the average von Neumann entropy of each fraction. The purpose is to show that the greatest fraction belongs to macrostates with maximum or near-maximum von Neumann entropy which, again, is consistent with a maximal entanglement between the system and the environment. However, since $\rho_{S}$, $F^{\rho_{S}}$, and $V_{SO(3)}^{\mathrm{norm}}$ are continuous functions of eigenvalues $\vec{\lambda }$, distinct macrostates are not well defined. To resolve this problem, we discretize S into discrete density operators, $\rho_{l}$, of equal area, and we discretize the range of $V_{SO(3)}^{\mathrm{norm}}$, L=[0,1], into discrete segments of equal length $L_{a}$. With these discretizations, $L_{a}$ represent the discrete macrostates in $H_{ES}$ to which fractions of S belong.

The proposed discretizations have two benefits. First, they allow us to identify $\rho_{l}$ with segments $L_{a}$ based on their volumes in $H_{ES}$ and compute
(35)$$S_{a}=\frac{|L_{a}|}{|\rho_{l}|}$$
where $|L_{a}|$ is the number of $\rho_{l}$ belonging to $L_{a}$, and $|\rho_{l}|$ is the total number of discrete density operators; this gives the fraction of S that belongs to each macrostate in $H_{ES}$. Second, they allow us to compute the average normalized von Neumann entropy of each $S_{a}$
(36)$$\bar{S_{VN}^{\mathrm{norm}}}\left(S_{a}\right)=\frac{\sum_{i=1}^{|L_{a}|}S_{VN}^{\mathrm{norm}}\left(\rho_{i}\right)}{|L_{a}|}$$
where $\rho_{i}$ belong to $L_{a}$. We then look at each $S_{a}$ and its $\bar{S_{VN}^{\mathrm{norm}}}\left(S_{a}\right)$ to see if feature (2) is demonstrated. Additionally, since $S_{VN}^{\mathrm{norm}}\left(\vec{\lambda }\right)$, $S_{L}^{\mathrm{norm}}\left(\vec{\lambda }\right)\in L$, we can compute Equations (35) and (36) for them as well, except we replace the volume with entropies when sorting $\rho_{l}$ into macrostates $L_{a}$. This allows us to compare them directly to $V_{SO(3)}^{\mathrm{norm}}$, which provides additional evidence that feature (2) is uniquely demonstrated by the ECG.

The probability simplex S is discretized into a finite $\rho_{l}$ of equal area by uniformly sampling it using the transformation
(37)$$\begin{array}{c} \lambda^{1}=1-\sqrt{\eta^{1}} \end{array}$$
(38)$$\begin{array}{c} \lambda^{2}=\sqrt{\eta^{1}}\left(1-\eta^{2}\right) \end{array}$$
(39)$$\begin{array}{c} \lambda^{3}=\sqrt{\eta^{1}}\eta^{2}, \end{array}$$
where $\eta^{1},\eta^{2}\in \left[0,1\right]$ are uniformly distributed in the unit interval, as seen in [29]. Dividing $\eta^{1}$ and $\eta^{2}$ into *ℓ* equal segments and transforming back to the $\vec{\lambda }$ basis divides S into $\ell^{2}$ discrete $\rho_{l}$, where $l\in [1,\ell^{2}]$; this is shown in Figure 5b.

The interval L=[0,1] is discretized by dividing it into *k* equal segments, $L_{a}$, where *a* is an integer between [1,k]; this is shown in Figure 5c. Given the discretization of S and *L*, one can compute Equations (35) and (36).

Choosing ℓ=300 and k=10, we compute $V_{SO(3)}^{\mathrm{norm}}$, $S_{L}^{\mathrm{norm}}$, and $S_{VN}^{\mathrm{norm}}$ at the center of squares in the $\vec{\eta }$ basis and assign that value to the corresponding $\rho_{l}$ in the $\vec{\lambda }$ basis. From Figure 5a, we see that the distance from the center of a given square is given by dl=1/(2ℓ). As *ℓ* goes to infinity, dl goes to zero, and the volume/entropies associated with the $\rho_{l}$ in the $\vec{\lambda }$ basis becomes more representative of the actual value at the center.

Coloring each $\rho_{l}$ using a color map derived from the volume and entropies assigned to them gives the first row of Figure 6. Notice how this simply produces the contour plots of Figure 4. To show the fraction of S associated with $L_{a}$, we assign an arbitrary color to each $L_{a}$ and color the $\rho_{l}$ in accordance with the $L_{a}$ in which they belong; this gives the second row of Figure 6. There is nothing special about the choice of colors; they are only meant to distinguish $L_{a}$. Computing Equation (35) and plotting the results gives the third row in Figure 6. Due to the triangular distortions of S by the transformation from $\vec{\eta }$ to $\vec{\lambda }$, these plots are produced with the restriction that $\eta^{1}\in \left(1/4,1\right]$ and $\eta^{2}\in \left(1/2,1\right]$. This guarantees the data in the analysis are within Weyl chambers [30] that do not include the triangular distortions (*The method for associating volume (or entropy) with a discrete density operator $\rho_{l}$ is only valid when $\rho_{l}$ is close to a regular polygon. Since the mapping from the $\vec{\eta }$ basis to the $\vec{\lambda }$ basis creates elongated triangles, the value of volume (or entropy) at the center is no longer representative of $\rho_{l}$. This can be seen in the second row of Figure 6 where the corner associated with the triangles is mono-colored while the corners consisting of more regular polygons have a clear gradient in color. The errors in counting which $\rho_{l}$ belong to which $L_{a}$ are ameliorated when triangular $\rho_{l}$ are not considered. And since S is symmetric, their removal does not affect the results*) of the grid in the $\vec{\lambda }$ basis. Finally, the fourth row of Figure 6 is given by Equation (36).

Looking at rows 3 and 4 of the first column of Figure 6, we see that over sixty percent of S consists of $\rho_{l}$ belonging to $L_{10}$. These are states for which $V_{SO(3)}^{\mathrm{norm}}\geq 0.9$. Furthermore, the average normalized von Neumann entropy of this class is 0.88 bits. This shows that the average entanglement entropy associated with $L_{10}$ is near maximal. These results are in stark contrast to the von Neumann and linear entropies whose $L_{10}$ segments make up less than thirty three percent of the total volume. This is significant because it shows that the von Neumann and linear entropies perform worse than the volume when reproducing feature (2) which is that most of the space of states consist of states near equilibrium. This suggests that the volume of the ECG uniquely captures features of a CG that is related to thermalization.

For Boltzmann’s original CG, over 99.99% of the γ-space consists of states at equilibrium. This is because it is assumed that one is working with a high-dimensional system with a number of particles on the order of Avogadro’s number. In this example, we are only working with three-level systems so the dimension of the space is vastly less. Nonetheless, we still show that the majority of $H_{ES}$ consists of states near equilibrium. In Section 3.4, we compute $\bar{S_{VN}^{\mathrm{norm}}}$ for states that occupy at least 99.99% of the volume of $H_{ES}$ and show that it tends toward one (maximum entanglement) as the dimension of the system increases.

### 3.4. Example: SO(N)

To extend the results from Section 3.3.2, we first provide an expression for $V_{SO(N)}^{\mathrm{norm}}$. We then use marginal density operators
(40)$$\rho_{S}\left(\lambda^{1}\right)=\lambda^{1}|\lambda^{1}\rangle \langle \lambda^{1}|+\frac{1-\lambda^{1}}{N-1}\sum_{i=2}^{N}|\lambda^{i}\rangle \langle \lambda^{i}|,$$
which are mixtures of a pure state and the maximally mixed state (of dimension N−1), to simplify the previous analysis for higher dimensions. This allows us to write $V_{SO(N)}^{\mathrm{norm}}$ as a function of $\lambda^{1}$. We then identify the value $\lambda^{1*}$ below which at least 99.99% of the volume exists. From here, the average normalized von Neumann entropy for $\rho_{S}\left(\lambda^{1}\right)$ between $\lambda^{1}\in \left[1/N,\lambda^{1*}\right]$ is computed. The purpose is to show that the average normalized von Neumann entropy for at least 99.99% of the coarse-grained space parameterized by $\lambda^{1}$ tends to one (maximal entanglement) as the dimension, *N*, of the system increases.

We compute the volume for SO(2)–SO(5) to construct $V_{SO(N)}$ by induction. The volume associated with SO(2) is computed by setting ψ=ξ=0 in Equation (17); this gives one metric component $dV_{SO(2)}=\sqrt{\lambda^{1}+\lambda^{2}}\;d\phi$. Inserting $dV_{SO(2)}$ into Equation (Equation 12) and integrating ϕ from zero to π/2 gives
(41)$$V_{SO(2)}=\left(\pi /2\right)\sqrt{\lambda^{1}+\lambda^{2}}=\pi /2.$$ This result is trivial and uninteresting since $\lambda^{1}+\lambda^{2}=1$, but it does provide necessary information for inferring the general form of $V_{SO(N)}$.

Although we have an analytical form of $dV_{SO(4)}$ produced by Mathematica, it cannot be simplified to a clean form as in Equations (22) and (32) when the number of parameters, $\vec{\xi }$, is greater than three (*A D×D matrix $g_{ij}$ has D! terms in the expansion of its determinant Det(g). SU(N)(SO(N)) has dimension $D=N^{2}-1\;\left(N\left(N-1\right)/2\right)$. Thus, SU(3) with D=8 has 8!=40,320 terms in Det(g) which we were unsuccessful in analytically simplifying in Mathematica. SO(3) with D=3 has 3!=6 terms in Det(g), while SO(4) with D=6 has 6!=720 terms, both of which can be simplified analytically*). To overcome this obstacle, we simplify $dV_{SO(4)}$ by setting $\vec{\xi }=0$. This is done because we notice that the volume elements $dV_{SO(3)}$, $dV_{SU(2)}$, and $dV_{SO(2)}$ are products between functions of λ’s and functions of $\vec{\xi }$, which may imply that volumes of the surfaces are product measures as seen in [30]. As such, the $\vec{\lambda }$ portion of the volume is removed from the integral, and the exact volume is merely scaled by factors of π. Assuming $dV_{SO(4)}$ is merely a product between a function of $\vec{\lambda }$ and cosines as in Equation (32), we set $\vec{\xi }=0$ to simplify it. Making this simplification gives
(42)$$dV_{{SO\left(N\right)|}_{\vec{\xi }=0}}=\prod_{i<j}^{N}\sqrt{\lambda^{i}+\lambda^{j}}\;d\xi_{1}d\xi_{2}\cdots d\xi_{N(N-1)/2}$$
where N=4 and N(N−1)/2 is the number of parameters of SO(N). Next, we justify the choice of setting $\vec{\xi }=0$ as valid by numerically computing $V_{SO(4)}$ directly, without setting $\vec{\xi }=0$, and compare it to Equation (42) for N=4.

Comparing the volumes given by Equation (42) with the direct numerical integration of $V_{SO(4)}$ where $\vec{\xi }\neq 0$ and the full integration over $\vec{\xi }$ is performed gives Figure 7.

This result numerically shows that Equation (42) (normalized to maximum) is a very good approximation of the actual normalized volume and that they may in fact be the same. This is not a proof, but it is a strong indication that the assumption leading to Equation (42) is valid. We also computed $dV_{SO(5)}$ and set $\vec{\xi }=0$ and obtained the same result for SO(4) which is that the volume, barring factors of π, is merely the square root of the product of all pairwise sums of eigenvalues. Using these results, along with $V_{SO(2)}$ and $V_{SO(3)}$, we infer by induction that
(43)$$V_{SO(N)}=\prod_{i<j}^{N}\sqrt{\lambda^{i}+\lambda^{j}}.$$ Now that we have a general form of $V_{SO(N)}$, we proceed with our procedure to extend the results from Section 3.3.2.

Inserting the choice of eigenvalues consistent with $\rho_{S}\left(\lambda^{1}\right)$ into Equation (43) and normalizing with respect to the maximum volume gives
(44)$$V_{SO(N)}^{\mathrm{norm}}\left(\lambda^{1}\right)=\frac{{\left(\lambda^{1}+\frac{1-\lambda^{1}}{N-1}\right)}^{\frac{N-1}{2}}{\left(2\frac{1-\lambda^{1}}{N-1}\right)}^{\frac{\left(N-1)(N-2\right)}{4}}}{{\left(\frac{2}{N}\right)}^{\frac{N(N-1)}{4}}}.$$ To show that the majority of $H_{ES}$ increasingly tends toward maximally entangled states (maximum von Neumann entropy of $\rho_{S}$), we plot Equation (44) for N=3,5,7,11, and 30 in Figure 8.

We see that the centroid of each plot tends toward states with maximum von Neumann entropy as *N* increases. To quantify these results, we identify the value $\lambda^{1*}$ for various values of *N* where $V_{SO(N)}^{\mathrm{norm}}\left(\lambda^{1*}\right)=10^{-4}$. For the values of *N* used, this choice of $\lambda^{1*}$ guarantees that
(45)$$\frac{\int_{1/N}^{\lambda^{1*}}V_{SO(N)}^{\mathrm{norm}}\left(\lambda^{1}\right)\;d\lambda^{1}}{\int_{1/N}^{1}V_{SO(N)}^{\mathrm{norm}}\left(\lambda^{1}\right)\;d\lambda^{1}}>0.9999,$$
where $\lambda^{1}=1/N$ indicates the maximally mixed $\rho_{S}\left(\lambda^{1}\right)$. Plotting the average normalized von Neumann entropy with $\lambda^{1}\in \left[1/N,\lambda^{1*}\right]$ as a function of *N* gives Figure 9. This clearly shows that the average normalized von Neumann entropy for at least 99.99% of $H_{ES}$ parameterized by $\lambda^{1}$ tends toward 1 as *N* becomes large. This implies that the vast majority of the coarse-grained space consists of equilibrium macrostates which are characterized by the maximum entanglement entropy.

From this analysis, we demonstrated feature (1) of Boltzmann’s CG for SU(2) and SO(3) by comparing them to the von Neumann and linear entropies in Figure 3 and Figure 4, respectively. We also demonstrated feature (2) for SO(3) by constructing an explicit CG and computing the average entanglement entropy of each macrostate and extended it to SO(N) using marginal density operators given by Equation (8). We did not include an analysis of SU(N) since computing the determinant of the metric becomes prohibitively difficult as the number of parameters, $\vec{\xi }$, increases.

## 4. Generalizing the Entanglement Coarse-Graining

In this section, we generalize our formalism to include unitary transformation of S in $P(H_{S})$. This allows us to define the metric components for SOI that belong to probability simplices with eigenbases rotated with respect to a fixed basis. Comparing density operators belonging to probability simplices with different eigenbases is a fundamental difference between classical and quantum fidelity measures. With this completed formalism, one could study quantum fidelity using a geometric approach provided by the SOI.

Given an orthonormal basis $\left\{|{\left(\lambda_{S}^{\rho }\right)}^{i}\rangle \right\}$ of $H_{S}$, all unitarily related orthonormal bases can be generated by
(46)$$\{|{\left(\lambda_{S}^{\sigma }\right)}^{i}\rangle \}=\{U_{S}|{\left(\lambda_{S}^{\rho }\right)}^{i}\rangle \}.$$ This gives the set of all unitarily related probability simplices $S^{\rho }$ and $S^{\sigma }$ in $P(H_{S})$ depicted in Figure 10. From here, the set of purifications associated with a density operator
(47)$$\sigma =\sum_{i=1}^{N}{\left(\lambda^{\sigma }\right)}^{i}|{\left(\lambda_{S}^{\sigma }\right)}^{i}\rangle \langle {\left(\lambda_{S}^{\sigma }\right)}^{i}|,$$
where ${\vec{\lambda }}^{\sigma }$ are free to be chosen independent of ${\vec{\lambda }}^{\rho }$, are given by (compare to Equation (Equation 7))
(48)$$|{\bar{\Gamma }}^{\sigma }\left(\vec{\xi }\right)\rangle =\left(U_{E}\left(\vec{\xi }\right)⊗\sqrt{\sigma }\right)|\Gamma_{ES}^{\sigma }\rangle$$
where (compare to Equation (Equation 6))
(49)$$|\Gamma_{ES}^{\sigma }\rangle =\sum_{i=1}^{N}|{\left(\lambda_{E}^{\sigma }\right)}^{i}\rangle |{\left(\lambda_{S}^{\sigma }\right)}^{i}\rangle .$$ Like Equation (Equation 6), $\left\{|{\left(\lambda_{E}^{\sigma }\right)}^{i}\rangle \right\}$ is a copy of $\left\{|{\left(\lambda_{S}^{\sigma }\right)}^{i}\rangle \right\}$ in $H_{E}$. Now, one simply inserts Equation (48) into Equation (Equation 11) to get the metric components of the surfaces of ignorance associated with $S^{\sigma }$.

This generalization may give new insights into quantum fidelity. The standard fidelity measure between arbitrary quantum states is the Uhlmann–Josza fidelity [31]. It has many equivalent definitions, two of which are given by
(50)$$\begin{array}{ccc} F_{UJ} & := & \max_{\left\{U_{S}\right\}}{\left|\mathrm{Tr}\left[\sqrt{\rho }\sqrt{\sigma }U_{S}^{T}\right]\right|}^{2} \end{array}$$
(51)$$\begin{array}{ccc} & \;\;\;\;\;\;\;\;\;\;\;\;\;\;\;= & \underset{\left\{{\vec{\xi }}_{\rho },{\vec{\xi }}_{\sigma }\right\}}{max}{\left|\langle {\bar{\Gamma }}^{\rho }\left({\vec{\xi }}_{\rho }\right)|{\bar{\Gamma }}^{\sigma }\left({\vec{\xi }}_{\sigma }\right)\rangle \right|}^{2} \end{array}$$
which are equations 9.110 and 9.97 in [1], respectively. If ρ and σ share the same eigenbasis, Equation (50) reduces to the classical fidelity between the eigenvalue spectrums of ρ and σ. This means that the difference between classical and quantum fidelity is the relationship between unitarily related eigenbases. Additionally, Equation (51) shows that the Uhlmann–Josza fidelity can also be understood as an optimization over the surfaces of ignorance. Therefore, the generalized ECG may provide new geometric insights into quantum fidelity as it relates to the ECG.

## 5. Discussion

In this paper, we introduced a new volume to quantify the amount of missing information or ignorance in a density operator $\rho_{S}$. This volume was computed by generating all purifications of $\rho_{S}$ and constructing the metric tensor associated with the manifold of purifications. We denoted these manifolds as surfaces of ignorance (SOI). The determinant of the metric provided a volume element which was integrated to compute the volume. Examples of the volume were provided for systems whose purifications were generated by Lie groups SU(2), SO(3), and SO(N). In these examples, the volumes were studied in the context of an entanglement-based quantum coarse-graining (CG) that we called the entanglement coarse-graining (ECG). This is a natural setting for studying the SOI since $\rho_{S}$ can be understood as the reduced density operator of a pure state thus making its von Neumann entropy the entanglement entropy between system *S* and environment *E*.

In the context of the ECG where the SOI are macrostates and purifications are microstates, we showed that our volumes captured two features of Boltzmann’s original CG. These features are essential to typicality arguments used to understand thermalization and the second law of thermodynamics. These features are: (1) a system beginning in an atypical macrostate of a smaller volume evolves to macrostates of a greater volume until it reaches the equilibrium macrostate, and (2) the equilibrium macrostate takes up the vast majority of the coarse-grained space especially as the dimension of the total system becomes large. Feature (1) was demonstrated by showing that the volume behaves like the von Neumann entropy in that it is zero on pure states, maximal on maximal mixed states, and is a concave function with respect to the purity of $\rho_{S}$. This was shown in Figure 3 and Figure 4 for the SU(2) and SO(3) examples, respectively. Feature (2) was demonstrated by Figure 6 for SO(3) and extended using SO(N) in Figure 8 and Figure 9.

The purpose of this work was not to study thermalization. Instead, we used information-based “thermalization” as a context to study our volumes in terms of the ECG. By demonstrating features (1) and (2) of the Boltzmann CG, we provided evidence that the intuitive understanding of the volume as a quantification of the missing information in $\rho_{S}$ was reasonable. Furthermore, it suggests that viewing these volumes as a multiplicity for an information/entanglement-based “thermalization” entropy constitutes a valid perspective. The ECG is also interesting in that it provides clear macro- and microstates for the entanglement entropy. Because of this, the equilibrium macrostate is consistent with a maximum entanglement between the *S* and *E*.

For future research, it would be interesting to study the well-known fact that most pure states of composite systems of high dimensions are close to maximally entangled [32] using the ECG. In the context of the ECG, this is simply an observation that the vast majority of the coarse-grained space of pure states consists of the equilibrium macrostate. This is feature (2) that was demonstrated in the examples of this paper and it is an essential feature of the results in [17,18,19,20,33]. It would also be interesting to study the relationship between the ECG and the analysis in [34], since the microstates of the ECG are envariant (entanglement-assisted invariant) states. Lastly, this research could be extended by defining a proper quantum Boltzmann entropy for the ECG. This is challenging since the volume goes to zero for pure states, which means simply taking the logarithm of the volume would result in a divergent entropy.

## Figures and Tables

**Figure 1 entropy-25-00788-f001:**
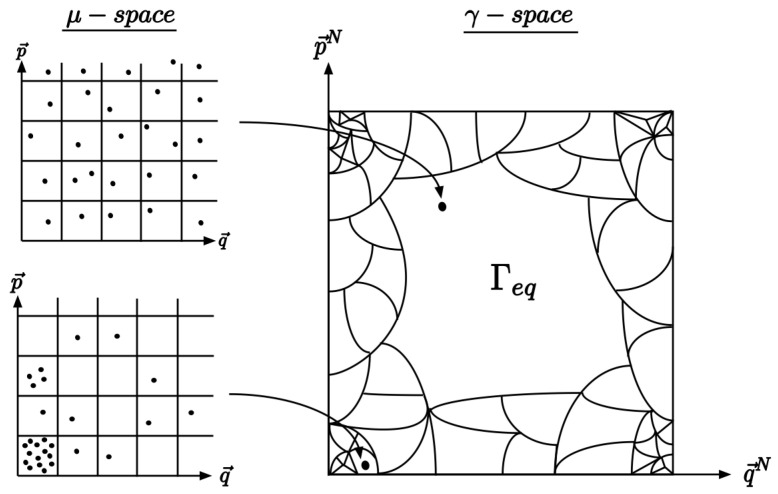
Illustration of Boltzmann’s original approach to coarse-graining inspired by Figure 2 in [14]. On the left are examples of distributions on the single particle phase space, the μ-space, while the right depicts the coarse-graining of the 6N-dimensional phase space, the γ-space. By dividing the μ-space into equal cells, macrostates are defined by simply counting the number of particles in each cell. Since each particle is indistinguishable, interchanging which particle occupies each cell does not change the macrostate; thus, there are many equivalent microstates for each macrostate. The size of each macrostate depends on the number of microstates it has. Boltzmann showed that distributions on the μ-space that are more uniform have more microstates, and the largest macrostate, $\Gamma_{eq}$, is associated with a gas in equilibrium.

**Figure 2 entropy-25-00788-f002:**
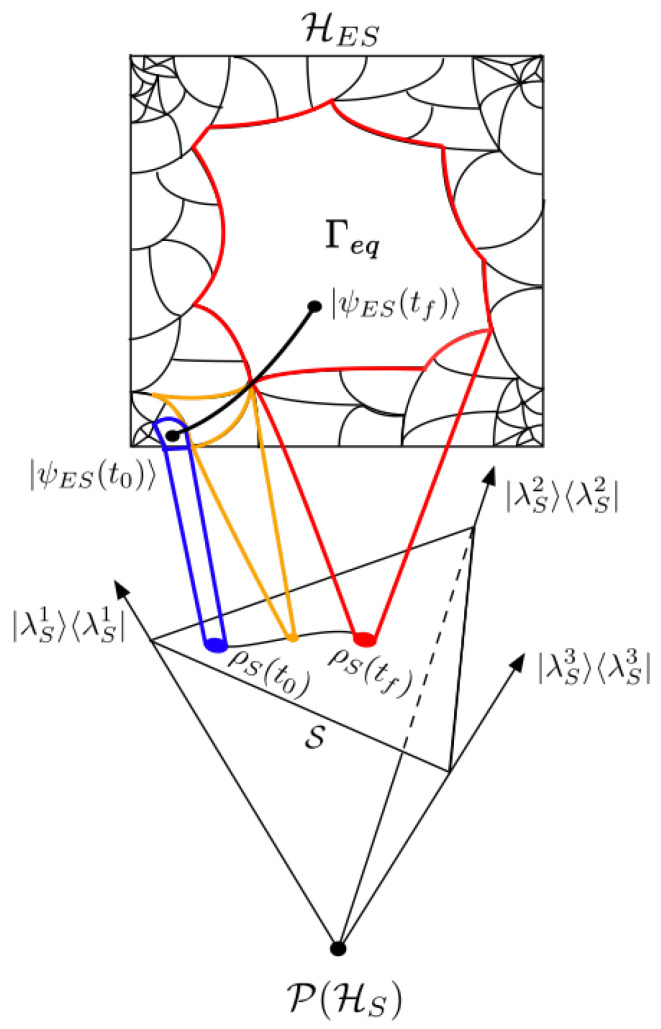
A conceptual example of an entangling process between $\rho_{S}$ and $\rho_{E}$. From the perspective of $\rho_{S}$, $|\psi_{ES}\rangle$ evolves from macrostates $F^{\rho_{S}}$ with a smaller volume to $F^{\rho_{S}}$ with a larger volume. If an observer only has access to the information in $\rho_{S}$, they cannot resolve the actual state of $|\psi_{ES}\rangle$ beyond the SOI depicted by the blue, orange, and red macrostates. For a global observer with access to $|\psi_{ES}\rangle$, the entangling process is a continuous curve of pure states from $|\psi_{ES}\left(t_{0}\right)\rangle$ to $|\psi_{ES}\left(t_{f}\right)\rangle$. This is the black curve in $H_{ES}$. Each $\rho_{S}\in S\subset P\left(H_{S}\right)$ has one unique $F^{\rho_{S}}\subset H_{ES}$. This implies a unique coarse-graining of S in $H_{ES}$.

**Figure 3 entropy-25-00788-f003:**
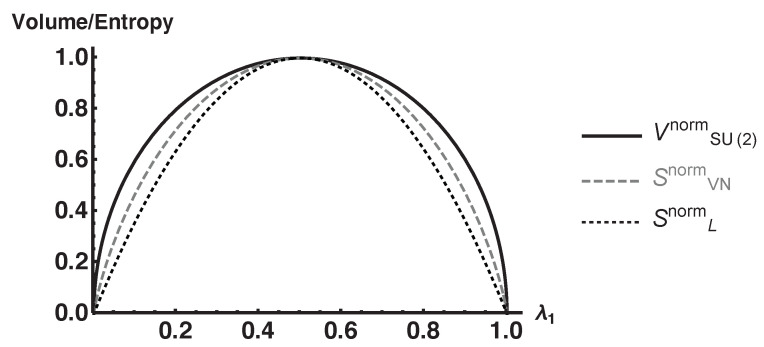
Plot of the normalized volume, von Neumann, and linear entropies for 2-level systems whose purifications are generated using SU(2).

**Figure 4 entropy-25-00788-f004:**
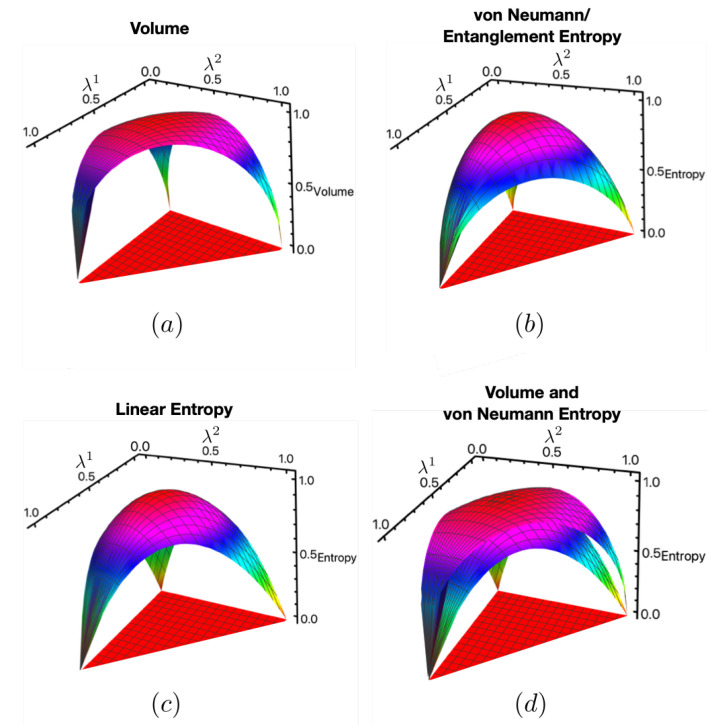
Comparison between the normalizations of $V_{SO(3)}$, von Neumann entropy, and linear entropy. This demonstrates that $V_{SO(3)}$ satisfies feature (1) of Boltzmann’s original CG for the example considered.

**Figure 5 entropy-25-00788-f005:**
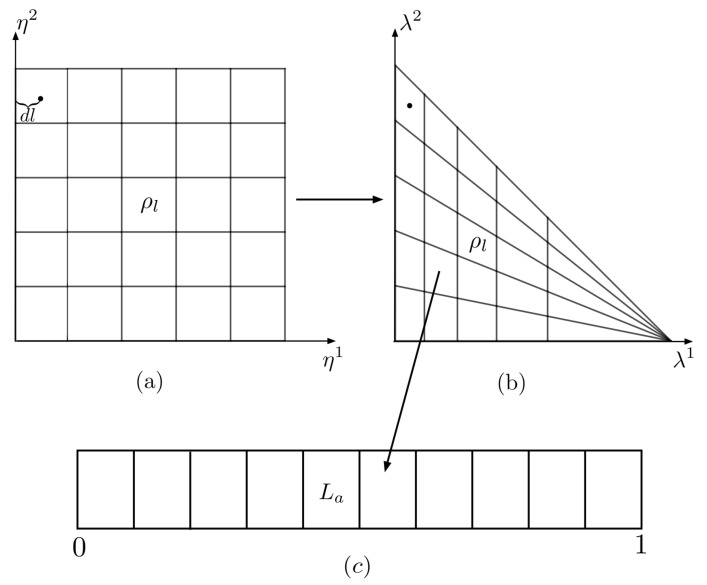
Discretization of the probability simplex S into a discrete $\rho_{l}$ of equal area, and the interval L=[0,1] into segments of equal length for ℓ=5 and k=10. In (**a**), we have the division of S in the $\vec{\eta }$ basis while (**b**) is that in the $\vec{\lambda }$ basis; the transformation is given by Equations (37)–(39). In (**c**), we have the sorting of $\rho_{l}$ into volume-equivalent classes $L_{a}$.

**Figure 6 entropy-25-00788-f006:**
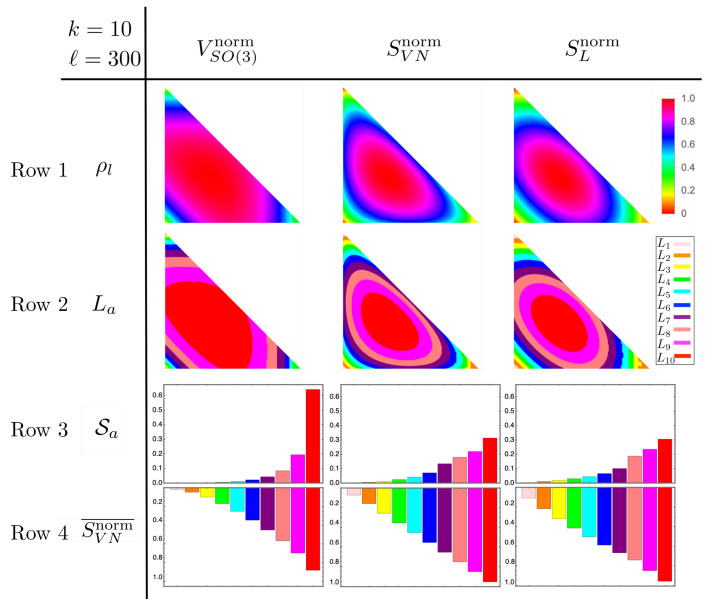
Results of coarse-graining $H_{ES}=R^{3}⊗R^{3}$. Row one is the discretization of S where each $\rho_{l}$ is colored using the volume or entropy of each column. Row two is the result of discretizing the interval L=[0,1] and sorting equivalent $\rho_{l}$ into segments $L_{a}$. Row three is the fraction of $\rho_{l}$ belonging to each $L_{a}$. Finally, row four is the average von Neumann entropy of each $L_{a}$. It should be noted that the data from the graphs do not include the triangular distortions caused by the discretization of S. We only used data from Weyl chambers that do not include triangles.

**Figure 7 entropy-25-00788-f007:**
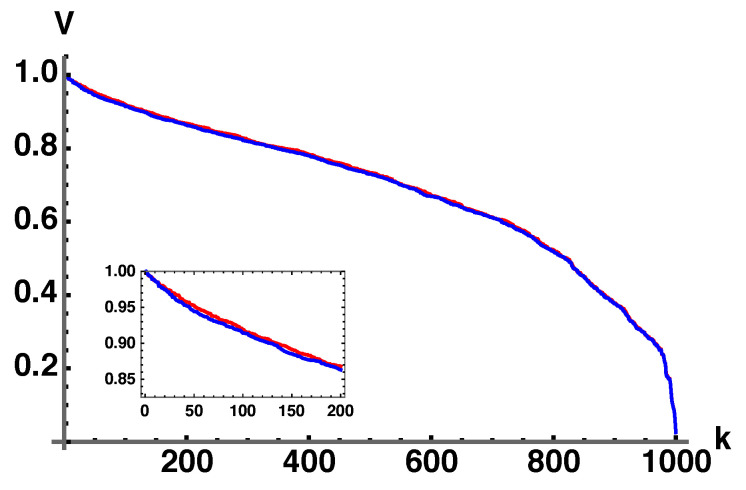
Plot comparing volumes given by Equation (42) with a direct numerical integration of $dV_{SO(4)}$. Both are normalized on their maximum values. To generate the plots, one thousand $\vec{\lambda }$’s were selected uniformly by generalizing Equations (37)–(39) to four dimensions and computing the corresponding volumes. The list of volumes and eigenvalues are sorted, k∈[1,1000], from largest to smallest. The red plot was computed from Equation (42), and the blue plot is a direct integration of $dV_{SO(4)}$ using a Monte Carlo integration. The inset is given to show that the plots are not exact but very close.

**Figure 8 entropy-25-00788-f008:**
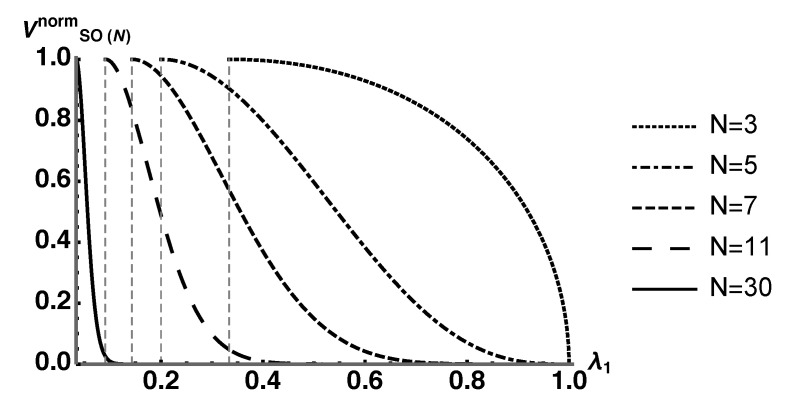
Plot of $V_{SO(N)}^{\mathrm{norm}}$ for N=3,5,7,11,30. The dashed vertical lines are located at the minimal value of $\lambda^{1}$ for each plot, which is 1/N, the maximally mixed state. Notice how the centroids tend toward maximally mixed states as pure states subsume less volume as *N* increases.

**Figure 9 entropy-25-00788-f009:**
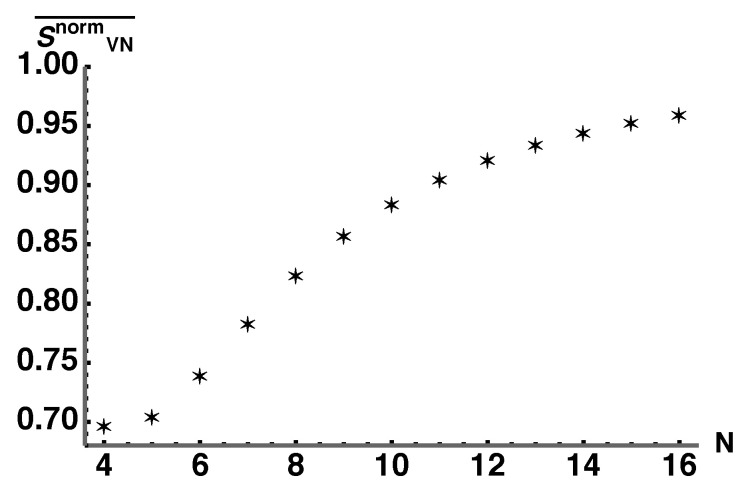
Plot of the average von Neumann entropy (normalized to the maximally mixed state) with $\lambda^{1}\in \left[1/N,\lambda^{1*}\right]$ as a function of *N*. This quantifies the results of Figure 8 by showing that the average von Neumann entropy of states whose volumes take over 99.99% of $H_{ES}$ tends toward 1 where 1 corresponds to the maximum entanglement entropy.

**Figure 10 entropy-25-00788-f010:**
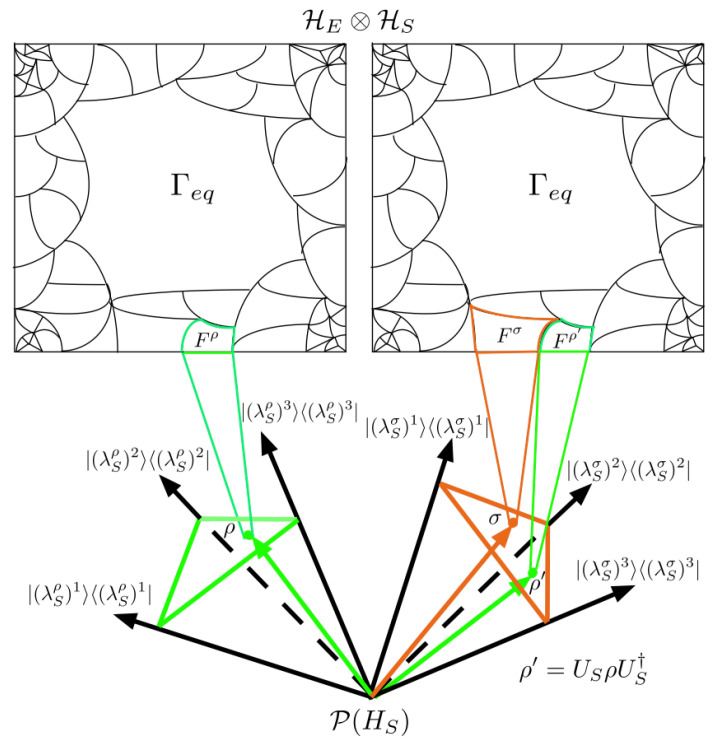
Depiction of generalized entanglement coarse-graining procedure to allow unitary transformations of S in $P(H_{S})$. The green simplex on the left associated with ρ is $S^{\rho }$, and the orange simplex on the right associated with σ is $S^{\sigma }$. The orthonormal basis of $S^{\sigma }$ is generated from unitary transformations $U_{S}$ applied to the orthonormal basis of $S^{\rho }$. Each simplex has a coarse-graining of $H_{ES}$ associated with it which is identical.

## Data Availability

Not applicable.

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
