# Peer review of "A Differential-Geometric Approach to Quantum Ignorance Consistent with Entropic Properties of Statistical Mechanics"

_entropy, 2023, doi:10.3390/e25050788_

Round 1

Reviewer 1 Report

The paper addresses an important issue of how to best quantify the concept of quantum ignorance (analogous to that of classical statistical theory). We assume that the von Neumann entropy of the reduced density matrix of the system (from the system environment entanglement) captures such notions. Authors define the concept of coarse-graining in quantum regimes to study the volume as the multiplicity of microstates [where microstates are the manifolds of purifications (surface of ignorance)]. They show that this volume quantifies the amount of ignorance in the reduced state of the system. They exemplify this concept for systems described by the representation, SU(2), SO(3) (analytically) and SO(N) (numerically). The feature they extract for the volume is analogous to that of von Neuman entropy, however, it shows that the equilibrium microstate (states with maximum volume) is typical when the dimension of coarse grain space becomes large.

I find that the study is adequate and qualifies for publication in its present form. I have one minor suggestion:

1. Eq. 25 can be written within the environment defined bellow

\begin{adjustwidth}{-\extralength}{0cm}

\end{adjustwidth},

then it might fit within the page.

Author Response

We took the referee's suggestion to use the adjustwidth environment in latex on Eq. 25 and it now fits on the page.  We thank the referee for the suggestion.

Reviewer 2 Report

The present paper provides a novel benchmark related to thermalizing quantum systems that is fundamental to understanding the nature of quantum systems. The authors show that the proposed metrics are effective in defining the process of thermalization for entangled macrostates in a system of interacting environments and systems. Their technique is a natural fit with Boltzmann’s original ideas, indicating the potential for further progress in this technique. Understanding the proposed metrics could provide new insights and a meaningful motivation for studying the thermalization of entanglement. Overall, the present manuscript is well-written, and the technical content is understandable. Thus, I recommend the publication of this manuscript after the following optional (minor) change.

1. If there are any limitations or conditions for applying the present metrics to other Lie groups, the authors could provide a more specific description.

Author Response

There are no additional restrictions that prevent us from applying our analysis to an arbitrary Lie group.  The only restriction is the difficulty in computing the determinant of the metric when the number of parameters is greater than 3. We gave a detailed explanation of this restriction in reference 32.   We thank the referee for their comments.